# Association of Long Non-Coding RNA Polymorphisms with Gastric Cancer and Atrophic Gastritis

**DOI:** 10.3390/genes11121505

**Published:** 2020-12-15

**Authors:** Vytenis Petkevicius, Greta Streleckiene, Kotryna Balciute, Alexander Link, Marcis Leja, Peter Malfertheiner, Jurgita Skieceviciene, Juozas Kupcinskas

**Affiliations:** 1Department of Gastroenterology, Lithuanian University of Health Sciences, 50161 Kaunas, Lithuania; juozas.kupcinskas@lsmuni.lt; 2Institute for Digestive Research, Lithuanian University of Health Sciences, 50161 Kaunas, Lithuania; greta.streleckiene@lsmuni.lt (G.S.); kotryna.balciute@lsmuni.lt (K.B.); jurgita.skieceviciene@lsmuni.lt (J.S.); 3Department of Gastroenterology, Hepatology and Infectious Diseases, Otto-von-Guericke University Magdeburg, 39120 Magdeburg, Germany; alexander.link@med.ovgu.de (A.L.); peter.malfertheiner@med.ovgu.de (P.M.); 4Institute for Clinical and Preventive Medicine, Faculty of Medicine, University of Latvia,1586 Riga, Latvia; marcis.leja@lu.lv; 5Faculty of Medicine, University of Latvia, 1586 Riga, Latvia; 6Department of Research, Riga East University Hospital, 1079 Riga, Latvia

**Keywords:** long non-coding RNA, single-nucleotide polymorphism, gastric cancer, atrophic gastritis

## Abstract

Long non-coding RNAs (lncRNA) play an important role in the carcinogenesis of various tumours, including gastric cancer. This study aimed to assess the associations of lncRNA *ANRIL, H19, MALAT1, MEG3, HOTAIR* single-nucleotide polymorphisms (SNPs) with gastric cancer and atrophic gastritis. SNPs were analyzed in 613 gastric cancer patients, 118 patients with atrophic gastritis and 476 controls from three tertiary centers in Germany, Lithuania and Latvia. Genomic DNA was extracted from peripheral blood leukocytes. SNPs were genotyped by the real-time polymerase chain reaction. Results showed that carriers of *MALAT1* rs3200401 CT genotype had the significantly higher odds of atrophic gastritis than those with CC genotype (OR-1.81; 95% CI 1.17–2.80, *p* = 0.0066). Higher odds of AG were found in a recessive model (CC vs. TT + CT) for *ANRIL* rs1333045 (OR-1.88; 95% CI 1.19–2.95, *p* = 0.0066). Carriers of *ANRIL* (rs17694493) GG genotype had higher odds of gastric cancer (OR-4.93; 95% CI 1.28–19.00) and atrophic gastritis (OR-5.11; 95% CI 1.10–23.80) compared with the CC genotype, and carriers of *HOTAIR* rs17840857 TG genotype had higher odds of atrophic gastritis (OR-1.61 95% CI 1.04–2.50) compared with the TT genotype; however, the ORs did not reach the adjusted significance threshold (*p* < 0.007). In summary, our data provide novel evidence for a possible link between lncRNA SNPs and premalignant condition of gastric cancer, suggesting the involvement of lncRNAs in gastric cancer development.

## 1. Introduction

Gastric cancer continues to be the third leading cause of cancer-related mortality worldwide, due to delayed diagnosis and poor prognosis [1,2]. The high mortality of gastric cancer is mainly associated with the lack of specific early symptoms and non-invasive diagnostic biomarkers. Existing methods for the diagnosis and therapy of gastric cancer remain inadequate; therefore, it is very important to explore new biomarkers and therapeutic targets for gastric cancer. In the past decade, the association of non-coding RNAs with cancer has been widely studied [3,4,5].

Non-coding RNAs have been classified as small non-coding RNAs, including microRNAs, and long non-coding RNAs (lncRNAs) [6]. The lncRNAs are more than 200 nucleotide size transcripts, which have been found in nearly all organisms, suggesting that they are a fundamental component of gene expression regulation [7]. Genomic studies have revealed a huge diversity of lncRNAs with different mechanisms of action [5,8]. The lncRNAs regulate gene expression at all levels, from modifying the epigenetic status of chromatin, to modulating the protein-coding mRNAs [9]. They contribute to many biological processes, such as cell proliferation, migration, development and differentiation [10]. Accumulating evidence suggests that dysregulation of lncRNAs results in tumour development, growth, invasion and metastasis [11,12].

The possible role of different genetic alterations in the development of gastric cancer has been studied [13,14,15]. Recent studies found that lncRNAs single-nucleotide polymorphisms (SNPs) may be involved in the carcinogenesis of gastric cancer [16,17]. An increasing amount of data demonstrate that gastric cancer patients have different lncRNA serum profiles compared with healthy controls [18]. These data indicate that lncRNAs may be promising gastric cancer biomarker and therapeutic targets [4,19,20]. However, little is known about associations of lncRNAs SNPs with neoplastic and preneoplastic (atrophic gastritis and intestinal metaplasia) conditions of the stomach.

This study aimed to examine associations of lncRNA *ANRIL* (rs17694493, rs1333045, rs1011970), *H19* (rs217727), *MALAT1* (rs3200401), *MEG3* (rs1054000), and *HOTAIR* (rs17840857) polymorphisms with gastric cancer and atrophic gastritis in European population.

## 2. Materials and Methods

### 2.1. Study Subjects

Study subjects were recruited at three gastroenterology centers in Lithuania (Department of Gastroenterology, Lithuanian University of Health Sciences, Kaunas), Latvia (Riga East University Hospital and Digestive Diseases Centre GASTRO, Riga) and Germany (Department of Gastroenterology, Hepatology and Infectious Diseases, Otto-von-Guericke University, Magdeburg). The controls and patients with atrophic gastritis were referred for upper endoscopy with biopsies from antrum because of dyspeptic symptoms and had no history of previous malignancy. Gastric cancer patients had histopathological verification of gastric adenocarcinoma. All individuals were of European descent.

In total, 1207 individuals were included in the study (476 controls, 118 atrophic gastritis patients, and 613 gastric cancer patients). There were 218 subjects from Germany, 620 from Latvia and 369 from Lithuania. The study was approved by Kaunas Regional Bioethics Committee (No. BE-2-10), Central Medical Ethics committee of Latvia (No. 01-29.1) and Institutional Review Board of Otto-von-Guericke University Magdeburg (No. 80/2011). Written informed consent was obtained from all subjects.

### 2.2. Selection of Long Non-Coding RNA SNPs

LncRNA were selected based on publications examining changes in the expression of these molecules in gastric cancer patients. All selected SNPs had rare allele frequency > 10% and fulfilled at least one of the following criteria: (1) SNP causes changes in RNAs secondary structure (*p* < 0.05; calculated using RNAsnp server [21]; (2) SNP causes gene expression changes in gastric tissue (*p* < 0.05; calculated using GTExPortal data [22]; (3) SNP value in RegulomeDB database ≤ 4 [23]. RegulomeDB annotates SNPs with known or expected regulatory elements (regions sensitive to DNAs, sites of transcription factors, promoter regions) located in the intergenic regions of the human genome [24]. The cut-off value of 4 was chosen to only select the variants that could be found in regulatory elements and thereby affect the binding of transcription factors. Finally, seven SNPs were selected (Table 1).

### 2.3. Genomic DNA Extraction and Genotyping

Genomic DNA was extracted from peripheral blood leukocytes using the salting-out method and stored at −20 °C until further processing. After DNA purification, the concentration and purity of each sample were measured using NanoDrop™ 2000 spectrophotometer. Genotyping was performed by using pre-designed TaqMan probes and primers (assay IDs: rs17694493-C__33349296_10; rs1333045-C___8766826_10; rs1011970-C___8766774_10; rs217727-C___2603707_10; rs3200401-C___3246069_10; rs17840857-C___3230519_20; rs1054000-C___7505737_10; Cat. No. 4351379; Applied Biosystems, Foster City, CA, USA) using Applied Biosystems 7500 Fast real-time PCR (RT-PCR) system and 7500 Software v2.3 for data analysis (Applied Biosystems International, Foster City, CA, USA). The reaction mixture was prepared according to the manufacturer’s protocol using TaqMan™ Universal Master Mix II, with UNG (Cat. No. 4440045; Applied Biosystems). Genotyping was performed according to the following protocol: 1 min, 60 °C pre-read; 2 min, 50 °C (for UNG activation) and 10 min, 95 °C for 1 cycle; 15 s, 95 °C and 1 min, 60 °C for 40 cycles; ended by 1 cycle 1 min, 60 °C post-read.

### 2.4. Statistical Analysis

The categorical variables were presented as proportions and compared using a χ2 test and Z-test with Bonferroni correction. Analysis of variance (ANOVA) was used to compare the mean age. In control group, each polymorphism was tested to ensure the fitting with Hardy–Weinberg equilibrium. Associations of gene polymorphisms with atrophic gastritis and gastric cancer were analyzed using multiple logistic regression analysis. The odds ratios (OR) were adjusted for sex, age and country of birth. For each polymorphism, four models were calculated: (1) each genotype was compared with the wild-type allele homozygous group; (2) recessive model (variant homozygous genotypes vs. heterozygotes and homozygotes for the wild-type allele); (3) dominant model (homozygotes variant + heterozygotes versus homozygotes for the wild-type allele); (4) variant allele vs. wild-type allele.

Statistical data analysis was performed using the statistical package IBM SPSS Statistics for Windows, Version 20.0 (Armonk, NY, USA: IBM Corp., released 2011). The Bonferroni-corrected α level was set at 0.007 (0.05/7 SNPs).

## 3. Results

The characteristics of the study population are presented in Table 2. Control subjects were younger than atrophic gastritis and gastric cancer patients and the proportion of men was considerably higher in gastric cancer patients than in atrophic gastritis patients and controls (Table 2). Therefore, to eliminate the potential bias of differences in age and sex distribution among the groups, these parameters were included in further logistic regression analysis.

The distributions of all analyzed SNPs and their genotypes in the control group did not differ from those predicted by a Hardy–Weinberg equilibrium: *p* = 0.863 for rs217727; *p* = 0.749 for rs3200401; *p* = 0.793 for rs17840857; *p* = 0.974 for rs1054000; *p* = 0.255 for rs17694493; *p* = 0.761 for rs1333045; *p* = 0.778 for rs1011970.

The prevalence of several genotypes differed comparing atrophic gastritis patients and controls (Table 3). The CT genotype of rs3200401 was more prevalent in atrophic gastritis patients than in controls (39.3% and 25.5%, respectively, *p* = 0.005), while the CC genotype was found more often in controls than in atrophic gastritis patients (70.5% and 59.8%, respectively, *p* = 0.031). Similar results were observed for TT genotype of rs17840857: 53.3% of controls and 39.8% of atrophic gastritis patients were carriers of this genotype (*p* = 0.007). The frequency of G allele was higher in AG patients (36.0%) compared with controls (27.4%) (*p* = 0.009). The CG genotype of rs17694493 was less prevalent in atrophic gastritis patients than in controls (16.9% and 25.9%, respectively, *p* = 0.012). A higher prevalence of CC genotype of rs1333045 was observed in atrophic gastritis patients compared with controls (33.6% and 21.9%, respectively, *p* = 0.014). No statistically significant differences in the prevalence of analyzed genotypes and allele frequencies were found between gastric cancer patients and controls (Table 3).

Logistic regression analysis showed a tendency of only one polymorphism of ANRIL lncRNA (rs17694493) to be associated with an increased risk of gastric cancer (Table 4). Carriers of the GG genotype had higher odds of gastric cancer when compared with the CC genotype (OR-4.93; 95% CI 1.28–19.00, *p* = 0.02); however, the difference did not reach the adjusted significance threshold. A similar tendency of the association was observed in a recessive model for rs17694493 (*ANRIL*), where comparison of GG vs. CC + CG genotypes showed a tendency of increased risk of gastric cancer (OR-5.02; 95% CI 1.31–19.28, *p* = 0.019). Statistically significant associations were found between some genotypes and atrophic gastritis risk. Carriers of CT genotype of rs3200401 (*MALAT1*) analyzed had higher odds of atrophic gastritis than those with CC genotypes (OR-1.81; 95% CI 1.17–2.80, *p* = 0.0066). The dominant model (comparison of TT + CT vs. CC genotypes) showed a tendency of increased risk of atrophic gastritis (OR-1.66; 95% CI 1.08–2.56, *p* = 0.020). A similar tendency of the association was observed for TG genotype of rs17840857 (*HOTAIR*). OR for atrophic gastritis was 1.61 (95% CI 1.04–2.50, *p* = 0.034), when the TG genotype was compared with the TT genotype. In a dominant model (GG + TG vs. TT), only the tendency of the association was found (OR-1.67; 95% CI 1.10–2.54, *p* = 0.016). The GG genotype of rs17694493 (*ANRIL*) could potentially be associated with higher odds of atrophic gastritis than the CC genotype (OR-5.11; 95% CI 1.10–23.80, *p* = 0.038); however, the difference did not reach the adjusted significance threshold. Heterozygous patients of rs17694493 (*ANRIL*) showed a tendency to have lower odds of atrophic gastritis than carriers of CC genotype. Significantly higher odds were found for a recessive model for rs1333045 (*ANRIL*), where comparison of CC vs. TT + CT genotypes showed an increased risk of atrophic gastritis (OR-1.88; 95% CI 1.19–2.95, *p* = 0.0066).

## 4. Discussion

Increasing evidence links dysregulation of lncRNAs to various diseases including cancers. The fact that lncRNAs are highly dysregulated in several types of cancer and are often expressed in a disease- and tissue-specific manner makes them an ideal biomarker candidate for cancer diagnosis. Our study analysed possible associations of several lncRNA SNPs with premalignant and malignant gastric conditions. LncRNAs investigated in the study were selected based on publications examining changes in the expression of these molecules in gastric cancer patients. Selected SNPs for four of the lncRNAs were: *ANRIL* (rs17694493 C > G, rs1333045 T > C, rs1011970 C >T), *H19* (rs217727 G > A), *MALAT1* (rs3200401 C > T), *MEG3* (rs1054000 A > C), *HOTAIR* (rs17840857 T > G). The study involved patients with atrophic gastritis and gastric cancer who were recruited at three gastroenterology centers in Lithuania, Latvia and Germany. We found that the CT genotype of rs3200401 in *MALAT1* gene and a recessive model for rs1333045 (*ANRIL*) were significantly associated with risk of atrophic gastritis. The polymorphisms rs17840857 in *HOTAIR* and rs17694493 in *ANRIL* showed a tendency to be associated with atrophic gastritis. Only one polymorphism rs17694493 in *ANRIL* gene showed a tendency to be linked to gastric cancer risk. To our knowledge, this is the first study reporting a possible link between lncRNA SNPs and the premalignant condition.

In our study, the association between SNP of *ANRIL* gene with atrophic gastritis reveals the possible involvement of this lncRNA SNP in gastric carcinogenesis. Antisense non-coding RNA in the INK4 locus (*ANRIL)* is a complex gene containing at least 21 exons. The best known molecular functions of *ANRIL* include gene regulation through chromatin modification complexes and influence over microRNA signalling pathways. Previous studies reported increased expression of ANRIL in various types of malignant tissue [25,26,27], including gastric cancer [28,29]. A study carried out in China found that the average level of ANRIL in gastric cancer tissues was significantly higher than those in corresponding non-tumour tissues. The high expression level of ANRIL in gastric cancer patients was associated with tumour size, advanced TNM stage and worse prognosis [29]. Moreover, genetic variants in *ANRIL* gene were associated with the risk of numerous conditions, including psoriasis [30], breast cancer [31] and multiple myeloma [32]. It was shown that SNP rs17694493 could be associated with prostate cancer [33]; however, this is the first study that showed associations of this SNP with atrophic gastritis and gastric cancer.

Metastasis-associated lung adenocarcinoma transcript 1 (MALAT1) is a biomarker for lung cancer metastasis and can govern hallmarks of lung cancer metastasis [34]. Wang et al. demonstrated that MALAT1 expression was significantly upregulated in gastric tumours compared with adjacent healthy tissue in patients with gastric cancer [35]. Furthermore, MALAT1 plasma expression was higher in patients with gastric cancer compared with healthy controls and had prognostic and diagnostic value [35]. Association studies of genetic variants in the *MALAT1* gene are widely studied and revealed a possible link with breast [36,37], colorectal [38,39] and hepatocellular [40] carcinogenesis. SNP rs3200401 particularly was previously investigated in breast cancer [36] and non-malignant conditions such as multiple sclerosis [41] and myocardial infarction [42]. No previous studies reported a significant association of this SNP in the risk of developing atrophic gastritis.

*HOX* Transcript Antisense RNA (HOTAIR) functions through an RNA product and serves as a scaffold to assemble regulators at the HOXD gene cluster, thereby promoting epigenetic repression of HOXD [43]. Overexpression of HOTAIR has been shown to promote metastasis, invasiveness of various types of cancer, including gastric cancer [44,45]. The studies have shown the association between high HOTAIR expression and poor survival in gastric cancer patients [45,46]. In addition to this, studies report many polymorphisms associated with risk of developing various malignancies including breast [47], cervical [48], lung [49] and gastric [50,51] cancers. To our knowledge, our study is the first in which rs17840857 was analyzed, and it showed that it could increase atrophic gastritis risk.

Previous studies have shown that H19 is upregulated in gastric cancer and that it is significantly correlated with poor prognosis of gastric cancer patients [52,53]. Furthermore, H19 has been shown to promote gastric cancer cell proliferation, invasion and metastasis through various mechanisms [53,54]. Polymorphisms in this gene were also shown to increase the risk for cancers such as hepatocellular [55], bladder [56] and gastric cancer [57]. We were not able to determine significant associations with *H19* SNPs and atrophic gastritis or gastric cancer.

Our study has certain limitations that have to be taken into account. The selection of lncRNAs is based on bioinformatical databases that may over- or underestimate real interaction effects. There were gender and age distribution differences between groups; however, when performing statistical analysis, we included gender and age as covariates, thus reducing the potential confounding effect.

In summary, our study found more possible associations between analyzed polymorphisms of lncRNA and atrophic gastritis than gastric cancer. It can be assumed that polymorphisms of *ANRIL* (rs1333045) and *MALAT1* (rs3200401) genes, as well as possibly *ANRIL* rs17694493 and *HOTAIR* rs17840857, are involved in the pathogenesis of gastric cancer. Future studies are needed to validate our findings in other cohorts.

## Figures and Tables

**Table 1 genes-11-01505-t001:** Selection criteria of long non-coding RNA SNPs.

Chr	Position (GRCh38)	SNP *	Alleles	Gene	MAF	RNAsnp **	GTExPortal ***	RegulomeDB
9	22041999	rs17694493	C > G	*ANRIL*	0.153	0.042	-	1b
9	22119196	rs1333045	T > C	*ANRIL*	0.442	0.621	-	3a
9	22062135	rs1011970	C >T	*ANRIL*	0.208	0.036	-	7
11	1995678	rs217727	G > A	*H19*	0.073	0.888	-	2b
11	65504361	rs3200401	C > T	*MALAT1*	0.213	0.886	4.50 × 10^−6^	4
12	53963973	rs17840857	T > G	*HOTAIR*	0.277	0.662	-	2b
14	100835645	rs1054000	A > C	*MEG3*	0.196	0.788	-	2b

Chr—chromosome, SNP—single nucleotide polymorphism; MAF—minor allele frequency; “-“ – no data, * dbSNP 2.0 Build 153; ** *p*-value showing the probability that SNP have an effect on local RNA secondary structure; *** *p*-value showing the probability that SNP affects gene expression in gastric tissue; Explanation of RegulomeDB estimates (data from GEO, ENCODE databases and publications): 1b—eQTL (locus of quantitative signs of expression) + transcription factor attachment + random motif + DNase interaction with DNA + DNase sensitive regions; 2b—transcription factor attachment + random motif + DNase interaction with DNA + DNase sensitive regions; 3a—transcription factor attachment + random motif + DNase sensitive regions; 4—transcription factor attachment + DNase sensitive regions; 5—transcription factor attachment or DNase sensitive regions; 7—no data.

**Table 2 genes-11-01505-t002:** Characteristics of the study population.

Characteristic	Controls *n* = 476	Atrophic Gastritis Patients *n* = 118	Gastric Cancer Patients *n* = 613	*p*-Value
Age (mean ± SD)	63.2 ± 9.4 *	67.9 ± 10.5	66.9 ± 11.1	< 0.001
Gender (%)				< 0.001
Male	30.5	35.6	64.3 **	
Female	69.5	64.4	35.7	

* *p* < 0.05 compared with atrophic gastritis and gastric cancer patients; ** *p* < 0.05 compared with controls and atrophic gastritis patients.

**Table 3 genes-11-01505-t003:** Genotype and allele frequencies in control group and atrophic gastritis and gastric cancer patients.

Genotype	Control		Atrophic Gastritis		Gastric Cancer		*p*-Value (Atrophic Gastritis and Control)	*p*-Value (Gastric Cancer and Control)
	n	%	n	%	n	%		
rs217727 *(H19)*								
GG	259	54.8	67	56.8	352	4.8		
AG	184	38.9	46	39	229	37.5	0.678	0.412
AA	30	6.3	5	4.2	29	57.7		
Allele G	702	74.2	180	76.3	933	76.5	0.515	0.224
Allele A	244	25.8	56	23.7	267	23.5		
rs3200401 *(MALAT1)*								
CC	335	70.5	70	59.8	416	3.5		
CT	126	25.5	46	39.3	171	28.1	0.014	0.73
TT	14	2.9	1	0.9	21	68.4		
Allele C	796	83.8	186	79.5	1003	82.5	0.117	0.421
Allele T	154	16.2	48	20.5	213	17.5		
rs17840857 *(HOTAIR)*								
TT	253	53.3	47	39.8	314	51.6		
TG	184	38.7	57	48.3	251	41.2	0.029	0.681
GG	38	8	14	11.9	44	7.2		
Allele T	690	72.6	151	64	879	72.2	0.009	0.811
Allele G	260	27.4	85	36	339	27.8		
rs1054000 *(MEG3)*								
AA	348	73.4	87	74.4	429	71.5		
AC	116	24.5	27	23.1	162	27	0.916	0.511
CC	10	2.1	3	2.6	9	1.5		
Allele A	812	85.7	201	85.9	1020	85	0.924	0.671
Allele C	136	14.3	33	14.1	180	15		
rs17694493 *(ANRIL)*								
CC	344	72.4	94	79.7	443	72.3		
CG	128	26.9	20	16.9	157	25.6	0.005	0.121
GG	3	0.6	4	3.4	13	2.1		
Allele C	816	85.9	208	88.1	1043	85.1	0.37	0.59
Allele G	134	14.1	28	11.9	183	14.9		
rs1333045 *(ANRIL)*								
TT	127	26.7	31	26.7	161	23.5		
CT	244	51.4	46	39.7	297	49.6	0.019	0.785
CC	104	21.9	39	33.6	141	26.9		
Allele T	498	52.4	108	46.6	619	51.7	0.109	0.729
Allele C	452	47.6	124	53.4	579	48.3		
rs1011970 *(ANRIL)*								
CC	353	74.3	91	77.1	463	76		
CT	111	23.4	24	20.3	139	22.8	0.779	0.313
TT	11	2.3	3	2.5	18	1.1		
Allele C	817	86	206	87.3	1065	87.4	0.607	0.326
Allele T	133	14	30	12.7	153	12.6		

The *p*-values < 0.05 are in bold.

**Table 4 genes-11-01505-t004:** Odds ratios of atrophic gastritis and gastric cancer by genotypes.

Genotype	Atrophic Gastritis			Gastric Cancer		
	OR	95% CI	*p*-Value	OR	95% CI	*p*-Value
rs217727 *(H19)*						
GG	1			1		
AG	0.96	0.63–1.48	0.867	0.89	0.67–1.17	0.388
AA	0.67	0.25–1.82	0.433	0.6	0.33–1.09	0.094
AA vs. AG + GG	0.68	0.26–1.82	0.444	0.63	0.36–1.13	0.123
AA + AG vs. GG	1.08	0.72–1.64	0.709	0.84	0.65–1.10	0.211
A vs. G	0.9	0.64–1.27	0.553	0.84	0.68–1.04	0.112
rs3200401 *(MALAT1)*						
CC	1			1		
CT	1.81	1.17–2.80	0.0066	1.11	0.82–1.50	0.495
TT	0.35	0.04–2.75	0.316	1.17	0.56–2.45	0.682
TT vs. CC + CT	0.29	0.04–2.24	0.234	1.13	0.54–2.37	0.738
TT + CT vs. CC	1.66	1.08–2.56	0.02	1.12	0.84–1.49	0.454
T vs. C	1.37	0.95–1.98	0.095	1.1	0.86–1.41	0.445
rs17840857 *(HOTAIR)*						
TT	1			1		
TG	1.61	1.04–2.50	0.034	1.22	0.92–1.60	0.169
GG	1.97	0.98–3.99	0.058	0.98	0.59–1.62	0.938
GG vs. TT + TG	1.57	0.81–3.05	0.186	0.9	0.55–1.47	0.676
GG + TG vs. TT	1.67	1.10–2.54	0.016	1.17	0.90–1.53	0.236
G vs. T	1.47	1.08–2.00	0.14	1.08	0.88–1.33	0.448
rs1054000 *(MEG3)*						
AA	1			1		
AC	0.94	0.58–1.54	0.81	1.12	0.83–1.52	0.451
CC	1.19	0.30–4.61	0.807	0.65	0.23–1.80	0.404
CC vs. AA + AC	1.2	0.31–4.66	0.79	0.63	0.23–1.74	0.37
CC + AC vs. AA	0.96	0.60–1.54	0.868	1.09	0.81–1.46	0.583
C vs. A	0.99	0.65–1.50	0.945	1.04	0.80–1.35	0.795
rs17694493 *(ANRIL)*						
CC	1			1		
CG	0.58	0.34–0.98	0.043	0.94	0.70–1.27	0.688
GG	5.11	1.10–23.80	0.038	4.93	1.28–19.0	0.02
GG vs. CC + CG	5.78	1.24–26.84	0.025	5.02	1.31–19.28	0.019
GG + CG vs. CC	0.68	0.41–1.12	0.13	1.01	0.75–1.36	0.938
G vs. C	0.83	0.54–1.30	0.417	1.09	0.84–1.43	0.508
rs1333045 *(ANRIL)*						
TT	1			1		
CT	0.77	0.46–1.28	0.313	0.96	0.70–1.32	0.816
CC	1.59	0.92–2.76	0.099	1.06	0.73–1.54	0.771
CC vs. TT + CT	1.88	1.19–2.95	0.0066	1.08	0.79–1.48	0.618
CC + CT vs. TT	1.01	0.63–1.61	0.976	0.99	0.74–1.34	0.956
C vs. T	1.28	0.96–1.72	0.095	1.03	0.85–1.24	0.793
rs1011970 *(ANRIL)*						
CC	1			1		
CT	0.8	0.48–1.32	0.38	1.02	0.75–1.40	0.889
TT	1.19	0.32–4.43	0.799	0.66	0.23–1.88	0.434
TT vs. CC + CT	1.25	0.34–4.64	0.741	0.65	0.23–1.87	0.427
TT + CT vs. CC	0.83	0.51–1.35	0.446	0.99	0.73–1.35	0.963
T vs. C	0.88	0.57–1.36	0.562	0.97	0.73–1.276	0.798

OR—odds ratio; CI—confidence interval.

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
