# Peer review of "Association of Long Non-Coding RNA Polymorphisms with Gastric Cancer and Atrophic Gastritis"

_genes, 2020, doi:10.3390/genes11121505_

Round 1
Reviewer 1 Report
The authors presented a study where they accessed the association of seven pre-selected SNPs within lncRNAs with gastric cancer and atrophic gastritis. Within the 7 selected SNPs, they found 2 of them show significant associations with atrophic gastritis, and 2 show potential associations with gastric cancer. The results are plain and straightforward and may provide some guidance for the community that is interested in studying those lncRNAs. However, I think several issues should be addressed to make the manuscript easier to understand.
Major issues:
1. As the authors mentioned in the discussion section, the selection of these 7 SNPs may somehow bias the effects observed. Therefore the authors should give clear explanations and justification of the criteria used to select these SNPs, which is not the case in the current form of the manuscript. For example in the Method section 2.2, what does "SNP value in RegulomeDB database <= 4" mean? The authors should explain what SNP value represents and why they would choose 4 as the cutoff.
Minor issues:
- This is related to major issue 1. In table 1 the authors should give more information on the SNPs, such as which dbSNP version did these SNP ids come from and the reference and alternative sequence of the SNPs (just like they mentioned in the discussion section, page 6, lines 167-168). Also, what are the values in the "RNAsnp" and "GTExPortal" columns? Why there are missing values in the "GTExPortal" columns? There is no way for the readers to figure out this in the current manuscript. Giving more clarification will make the table more self-explanatory.
- The abbreviations of gastric cancer (GC) and atrophic gastritis (AG) are sometimes a bit confused with genotypes. The authors should consider using the full names of the diseases whenever they refer to them in a sentence describing the genotypes.
- In table 4, the rs217727 genotypes and the values on the right are misaligned.
Author Response
Response to Reviewer 1 comments
Thank you very much for reviewing our manuscript. We appreciate your comments and suggestions. We hope that we have successfully addressed all of the concerns raised, and we believe that the manuscript has been substantially improved. Our detailed responses to the comments and the description of the changes we have made to the manuscript are provided below.
Point 1: As the authors mentioned in the discussion section, the selection of these 7 SNPs may somehow bias the effects observed. Therefore the authors should give clear explanations and justification of the criteria used to select these SNPs, which is not the case in the current form of the manuscript. For example in the Method section 2.2, what does "SNP value in RegulomeDB database <= 4" mean? The authors should explain what SNP value represents and why they would choose 4 as the cutoff.
Response 1: In response to reviewer‘s comments, we would like to explain that RegulomeDB is a database that provides functional annotation of SNPs based on data of many sources such as the ENCODE Project Consortium (2012), NCBI Sequence Read Archive, etc. Variants can be classified into one of four RegulomeDB categories with scores ranging from 1 to 6 indicating putative functions (SNP value in RegulomeDB database): (1) Likely to affect binding and linked to the expression of a gene target; (2) Likely to affect binding; (3) Less likely to affect binding; (4) Minimal binding evidence. (Table 2 in Rosenthal SL, Barmada MM, Wang X, Demirci FY, Kamboh MI. Connecting the dots: potential of data integration to identify regulatory SNPs in late-onset Alzheimer's disease GWAS findings. PLoS One. 2014 Apr 17;9(4):e95152. doi: 10.1371/journal.pone.0095152. PMID: 24743338).
We added this sentence to the Method section (line 87): ‘The cut-off value of 4 was chosen to only select the variants that could be found in regulatory elements and thereby affect the binding of transcription factors.’
Point 2: This is related to major issue 1. In table 1 the authors should give more information on the SNPs, such as which dbSNP version did these SNP ids come from and the reference and alternative sequence of the SNPs (just like they mentioned in the discussion section, page 6, lines 167-168). Also, what are the values in the "RNAsnp" and "GTExPortal" columns? Why there are missing values in the "GTExPortal" columns? There is no way for the readers to figure out this in the current manuscript. Giving more clarification will make the table more self-explanatory.
Response 2: Following reviewer’s comments, version and build number of dbSNP are now provided in the footer of Table 1. Reference and alternative alleles are also added to the Table 1. RNAsnp enables to predict SNP effect on local RNA secondary structure and the value reported is a p-value. GTExPortal column shows the probability that the variant affects the expression of a gene in tissue-specific manner (in our case gastric tissue). Missing values means that there is no data on gastric tissue. The explanations of “RNAsnp” and “GTExPortal” columns are now added in the footer of Table 1.
Point 3: The abbreviations of gastric cancer (GC) and atrophic gastritis (AG) are sometimes a bit confused with genotypes. The authors should consider using the full names of the diseases whenever they refer to them in a sentence describing the genotypes.
Response 3: As the reviewer suggested, the abbreviations of gastric cancer (GC) and atrophic gastritis (AG) were changed by full names of the diseases.
Point 4: In table 4, the rs217727 genotypes and the values on the right are misaligned.
Response 4: Corrected.
Sincerely,
On behalf of the authors,
Vytenis Petkevicius
Department of Gastroenterology, Lithuanian University of Health Sciences

Reviewer 2 Report
The authors showed some polymorphisms of lncRNA are highly detectable in preneoplasmic condition, including AG, as well as GC. The lncRNAs are highlighted as one of cancer regulatory RNA groups, thus the concept of this manuscript is interesting.
However, the selection of lncRNA is not adequately.
Major point
- Genome wide association study has already been conducted in gastric cancer in other ethnic group, such as Chinese, and some polymorphisms in lncRNAs identified the cancer related polymorphisms. Authors should test these polymorphisms in AG as well as lncRNAs presented in this manuscript.
- Case control study, such as survival rate, should be performed to assess whether identified polymorphism in GC is correlated with prognosis.
Minor point
- Table3, add marks that represent significant differences for ease to understand the data.
- Table3, each p-value (the ration of control vs AG/GC) should be shown.
- Table4, the position of number of OR, 95% CI and P-value in rs217727 is correct?
Author Response
Response to Reviewer 2 comments
We would like to thank you for reviewing our manuscript. We hope that we have successfully addressed all of the concerns raised. Our detailed responses to the comments and the description of changes we have made to the manuscript are provided below.
Point 1: Genome wide association study has already been conducted in gastric cancer in other ethnic group, such as Chinese, and some polymorphisms in lncRNAs identified the cancer related polymorphisms. Authors should test these polymorphisms in AG as well as lncRNAs presented in this manuscript.
Response 1: We agree that there are other SNPs in lncRNAs that might be associated with gastric cancer risk. We cited several studies carried out in China, which showed that polymorphisms in HOTAIR and H19 genes might increase the risk of gastric cancer (references 50, 51, 57). Criteria of the selection of SNPs analysed in our manuscript was described in the Methods section (SNP causes changes in RNAs secondary structure; SNP causes gene expression changes in gastric tissue; SNP value in RegulomeDB database ≤4). Associations of all selected SNPs with gastric cancer as well as atrophic gastritis were analysed.
Point 2: Case control study, such as survival rate, should be performed to assess whether identified polymorphism in GC is correlated with prognosis.
Response 2: We agree that analysis of the survival rate of gastric cancer patients with different lncRNAs polymorphisms would be interesting; however, we do not have survival data of our patients.
Point 3: Table3, add marks that represent significant differences for ease to understand the data.
Response 3: The P values <0.05 are now marked in bold.
Point 4: Table3, each p-value (the ration of control vs AG/GC) should be shown.
Response 4: The distribution of gene polymorphisms in control group versus atrophic gastritis group as well as in control group versus gastric cancer group was compared using a χ2 test. P values for control group vs atrophic gastritis group are presented in 8th column of Table 3 and P values for control group vs gastric cancer group are shown in the 9th column of Table 3.
Point 5: Table4, the position of number of OR, 95% CI and P-value in rs217727 is correct?
Response 5: We confirm that position of all numbers in Table 4 are correct.
Sincerely,
On behalf of the authors
Vytenis Petkevicius
Department of Gastroenterology, Lithuanian University of Health Sciences

Reviewer 3 Report
The manuscript by Petkevicius et al. "Association of long non-coding RNA polymorphisms with gastric cancer and atrophic gastritis" demonstrates an association between lncRNA SNPs and gastric cancer development. The content of the publication enriches the existing knowledge in the field and is suitable for publication after minor revision. I kindly ask the authors to correct/clarify the following points:
Abstract:
Line 21: Correct "centres" to centers.
1. Introduction:
This study is performed on 3 cohorts - gastric cancer, atrophic gastritis, and controls. It might be, however, unclear, that AG is a precursor condition considering it is not mentioned in the introduction. I strongly recommend adding an explanatory sentence into this section, that will clarify later inclusion of AG into the study.
Specific comments:
Line 36: Correct to cancer-related.
Line 53: Reference no. 17 doesn't discuss SNPs in the carcinogenesis of GC, please delete or replace this reference.
2. Materials and Methods
2.2
How many SNPs were selected from the publications in total?
Specific comments:
Table 1.: In the table, you use the position of SNPs from the archive GRCh37. Since there is an improved archive, I recommend stating the position according to GRCh38.
2.3.
More information about PCR would strengthen this report.
What Master Mix/chemistry was used, please state the product number?
If Taq-Man probes and primers were pre-designed, please state the product number.
3. Results
In general, I found it unfortunate that the abbreviation GC and AG match the designation of the nitrogenous base. In the continuous long text, it might be confusing for the reader. If used for disease, please state the whole name in parentheses.
For better clarity in Table 3. and Table 4. suggest providing next to the SNP code also the name of the lncRNA.
Specific comments:
Table 3.: rs3200401, P-value (GC and control) - correct the decimal separator, comma instead of the dot.
Line 143: Incorrectly stated genotype (fix to GG vs CC + CG).
4. Discussion
Specific comments:
Line 176: Try "involvement" instead of "application".
Line 209: Add "cancer" after gastric.
Author Response
Response to Reviewer 3 comments
We would like to thank you for your efforts and time to read and revise our manuscript. We appreciate your comments and suggestions that helped us to improve our manuscript. Our detailed responses to the comments and the description of the changes we have made to the manuscript are provided below.
Point 1: Line 21: Correct "centres" to centers.
Response 1: Corrected.
Point 2: This study is performed on 3 cohorts - gastric cancer, atrophic gastritis, and controls. It might be, however, unclear, that AG is a precursor condition considering it is not mentioned in the introduction. I strongly recommend adding an explanatory sentence into this section, that will clarify later inclusion of AG into the study.
Response 2: Following reviewer’s advice, we added suggested explanation in the introduction (line 59). ‘However, little is known about associations of lncRNAs SNPs with neoplastic and preneoplastic (atrophic gastritis and intestinal metaplasia) conditions of the stomach’.
Point 3: Line 36: Correct to cancer-related.
Response 3: Corrected
Point 4: Reference no. 17 doesn't discuss SNPs in the carcinogenesis of GC, please delete or replace this reference.
Response 4: Thank you for noticing our mistake. We have changed the reference no. 17.
Point 5: How many SNPs were selected from the publications in total?
Response 5: Criteria of the selection of SNPs analysed in our manuscript was described in the Methods section (SNP causes changes in RNAs secondary structure; SNP causes gene expression changes in gastric tissue; SNP value in RegulomeDB database ≤4). Only lncRNA were selected based on publications examining changes in the expression of these molecules in gastric cancer patients.
Point 6: In the table, you use the position of SNPs from the archive GRCh37. Since there is an improved archive, I recommend stating the position according to GRCh38.
Response 6: Positions are now corrected accordingly.
Point 7: More information about PCR would strengthen this report. What Master Mix/chemistry was used, please state the product number? If Taq-Man probes and primers were pre-designed, please state the product number.
Response 7: As the reviewer recommended, more information on DNA genotyping was added (lines 103-113): Genotyping was performed by using pre-designed TaqMan probes and primers (assay IDs: rs17694493 - C__33349296_10; rs1333045 - C___8766826_10; rs1011970 - C___8766774_10; rs217727 - C___2603707_10; rs3200401 - C___3246069_10; rs17840857 - C___3230519_20; rs1054000 - C___7505737_10; Cat. No. 4351379; Applied Biosystems, USA) using Applied Biosystems 7500 Fast real-time PCR (RT-PCR) system and 7500 Software v2.3 for data analysis (Applied Biosystems International, USA). The reaction mixture was prepared according to the manufacturer’s protocol using TaqMan™ Universal Master Mix II, with UNG (Cat. No. 4440045; Applied Biosystems, USA).
Point 8: In general, I found it unfortunate that the abbreviation GC and AG match the designation of the nitrogenous base. In the continuous long text, it might be confusing for the reader. If used for disease, please state the whole name in parentheses.
Response 8: Following reviewer’s suggestion, the abbreviations of gastric cancer (GC) and atrophic gastritis (AG) were changed by full names of the diseases.
Point 9: For better clarity in Table 3. and Table 4. suggest providing next to the SNP code also the name of the lncRNA.
Response 9: The names of the lncRNAs were added in Table 3 and Table 4.
Point 10: Table 3.: rs3200401, P-value (GC and control) - correct the decimal separator, comma instead of the dot.
Response 10: Corrected.
Point 11: Line 143: Incorrectly stated genotype (fix to GG vs CC + CG).
Response 11: We apologize for our mistake. It was corrected.
Point 12: Line 176: Try "involvement" instead of "application".
Response 12: Corrected.
Point 13: Line 209: Add "cancer" after gastric.
Response 13: Corrected.
Sincerely,
On behalf of the authors,
Vytenis Petkevicius
Lithuanian University of Health Sciences

Round 2
Reviewer 2 Report
The revised version of manuscript is improved. However, there is a comment to be addressed prior to publication.
Following is my specific comment.
- Through the whole of manuscript, “association” is overstated.
Authors identified “the polymorphisms of lncRNAs” which are detectable in gastric cancer or atrophic gastritis.
However, “the association between polymorphism and gastric cancer or atrophic gastritis” has been unknown.
Authors should tone down.

Author Response
Response to Reviewer 2 comments (2 round)
We would like to thank you once again for reading our manuscript. We appreciate your comments, although we cannot fully agree with them. Our responses to the comments are provided below.
Point 1: Through the whole of manuscript, “association” is overstated. Authors identified “the polymorphisms of lncRNAs” which are detectable in gastric cancer or atrophic gastritis. However, “the association between polymorphism and gastric cancer or atrophic gastritis” has been unknown. Authors should tone down.
Response 1: We believe that this comment came because we understand a little differently the word ‘association.’ We would like to cite the definition of genetic association studies with which we fully agree: ‘Genetic association studies test for a correlation between disease status and genetic variation to identify candidate genes or genome regions that contribute to a specific disease. A higher frequency of a single-nucleotide polymorphism (SNP) allele or genotype in a series of individuals affected with a disease can be interpreted as meaning that the tested variant increases the risk of a specific disease’
(http://cshprotocols.cshlp.org/content/2012/3/pdb.top068163.full)
According to this definition, our study aimed to analyze the association of lncRNA single nucleotide polymorphisms (SNPs) with gastric cancer and atrophic gastritis. Furthermore, we used logistic regression for data analysis. This method of statistical analysis evaluates the association between risk factor and diseases. We do not claim that established associations are necessarily causal. The explanation of the potential mechanisms of the associations between analyzed SNPs and cancer are provided in the discussion section.
On the other hand, we agree that the word ‘association’ is used in our manuscript too often. So, we changed the word ‘association’ to ‘link’ or ‘results’ in some places. Also, we added the word ‘possible’ to ‘association’ in some places to emphasize that the defined association have to be proofed in the studies in other cohorts. The changes are made in lines 32,146,184,195 and 238.
Sincerely,
On behalf of the authors
Vytenis Petkevicius
Department of Gastroenterology, Lithuanian University of Health Sciences